

# Evidence of positive selection and a novel phylogeny among five subspecies of song sparrow (*Melospiza melodia*) in Alaska

Caitlyn C. Oliver Brown[1,2,3], Keiler A. Collier[1,2,4], David J. X. Tan[3], Kendall Mills[1,2], Fern Spaulding[1,2], Travis C. Glenn[5], Christin Pruett[6] and Kevin Winker[1,2]

[1] University of Alaska Museum, Fairbanks, Alaska, United States
[2] Department of Biology and Wildlife, University of Alaska Fairbanks, Fairbanks, Alaska, United States
[3] Department of Biology, University of New Mexico, Albuquerque, New Mexico, United States
[4] RENECO International Wildlife Consultants, Abu Dhabi, United Arab Emirates
[5] Department of Environmental Health Science and Institute of Bioinformatics, University of Georgia, Athens, Georgia, United States
[6] Department of Biology, Ouachita Baptist University, Arkadelphia, Arkansas, United States

Corresponding author
Caitlyn C. Oliver Brown,
caitlyn.brown@gmail.com

## ABSTRACT

Local adaptation occurs when populations evolve traits in response to local environmental challenges. Isolated island populations often experience different selection pressures than their mainland counterparts, which enables the study of how phenotypes and genotypes respond to differing selection regimes. We studied a group of five phenotypically differentiated subspecies of song sparrow (*Melospiza melodia*) in Alaska that demonstrate striking body size, color, and migratory behavioral differences to examine the effects of local adaptation on phenotypes and genotypes. We examined the phenotypic attributes of these populations and used whole-genome data to determine relationships and test candidate loci for evidence of selection. Phenotypic measurements of museum specimens ($n = 227$) quantified the dramatic size differences among these populations, with westernmost *M. m. maxima* being ~1.6 times larger than easternmost *M. m. rufina*. Using ultraconserved elements (UCEs) and McDonald-Kreitman tests, we showed that seven candidate genes associated with bill size, circadian rhythm regulation, plumage color, and salt tolerance exhibited signs of putative positive selection. Phylogenetic analysis of UCEs identified *M. m. maxima* as sister to the other Alaska *M. melodia* subspecies. This suggests *M. m. maxima* colonized earliest, perhaps before the last glacial maximum, and that Alaska was later recolonized by ancestors of the remaining four subspecies.

## INTRODUCTION

Our current understanding of evolution suggests that adaptation to local conditions enables populations to persist and that geographic variation among populations often reflects these processes. Local adaptation is defined as the process by which a local population evolves traits best suited for its environment (*Turesson, 1922*; *Mayr, 1963*;

*Williams, 1966*; *Kawecki & Ebert, 2004*; *Savolainen, Lascoux & Merilä, 2013*). Local adaptation to a specific environment can potentially lead to speciation (*Savolainen, Lascoux & Merilä, 2013*; *Tiffin & Ross-Ibarra, 2014*), but this process of adaptation can be hindered by the homogenizing effects of dispersal and gene flow (*Kawecki & Ebert, 2004*; *Savolainen, Lascoux & Merilä, 2013*). While it is still challenging to identify the genetic basis of local adaptation, the advent of high-throughput genomic data can be used to identify potential genes under selection (*Tiffin & Ross-Ibarra, 2014*; *Bomblies & Peichel, 2022*). In one method for examining local adaptation, candidate genes are chosen *a priori* due to their putative phenotypic effects of interest and adaptive potential can be tested (*e.g.*, *Walsh et al., 2012*). Examining local adaptation has been done in many terrestrial organisms, such as lizards (*Losos, Warheitt & Schoener, 1997*), birds (*Grant, 1981*), spiders (*Gillespie, 2002*), and plants (*Choi et al., 2021*).

Island systems have been influential in our understanding of the processes of evolution (*MacArthur & Wilson, 1967*; *Losos & Ricklefs, 2009*; *Warren et al., 2015*). Islands, especially archipelagos, are natural laboratories for studying evolution, given their relatively young geological age and geographic isolation from the mainland (*Losos & Ricklefs, 2009*; *Warren et al., 2015*). Because of these conditions, island populations provide a way to determine how local adaptation accrues. The majority of research on local adaptation has occurred on tropical or mid-latitude islands, however there has been a recent focus on high-latitude island systems (*Whittaker & Fernadez-Palacios, 2007*) such as the Aleutian Islands in the North Pacific Ocean, a volcanic archipelago that extends ~1,800 km from western Alaska towards eastern Russia (*Murie, 1959*). During the Pleistocene, Alaska experienced multiple cycles of glaciation and glaciers covered much of the North American continent. However, ice-free zones, or refugia, were present that could have allowed populations to persist, and the Aleutian Islands are hypothesized to contain some of these refugia (*Pruett & Winker, 2005a*; *Winker et al., 2023*). Isolated avian populations on these islands often exhibit both lower genetic diversity than mainland counterparts and phenotypic differences, such as larger body size (*Pruett, Li & Winker, 2018*). One such species is the song sparrow (*Melospiza melodia*, Passeriformes: Passerellidae), a widely distributed songbird found only in North America with ~25 recognized subspecies (*Aldrich, 1984*; *Patten & Pruett, 2009*). These subspecies exhibit a wide range of phenotypes and life histories and reside in a variety of environments across North America (*Aldrich, 1984*; *Patten & Pruett, 2009*; *Arcese et al., 2020*; *Carbeck et al., 2023*). Because of this, song sparrows are an excellent candidate for the study of local adaptation.

Five song sparrow subspecies occur across southern Alaska, from the western Aleutian Islands to southeast Alaska. These subspecies, from west to east, are *M. m. maxima, M. m. sanaka, M. m. insignis, M. m. caurina*, and *M. m. rufina* (Fig. 1; *Gibson & Withrow, 2015*). Prior genetic research suggested that ancestral song sparrows colonized the Aleutian Islands sequentially, from east to west (*Pruett & Winker, 2005b*). Across this complex, these subspecies exhibit different life history traits, presumably due to differences in their environments. The subspecies that occur furthest west (*maxima* and *sanaka*) are non-migratory residents on the Aleutian Islands, a relatively harsh maritime environment. These two subspecies are found in grassy slopes along marine beaches, a habitat considered
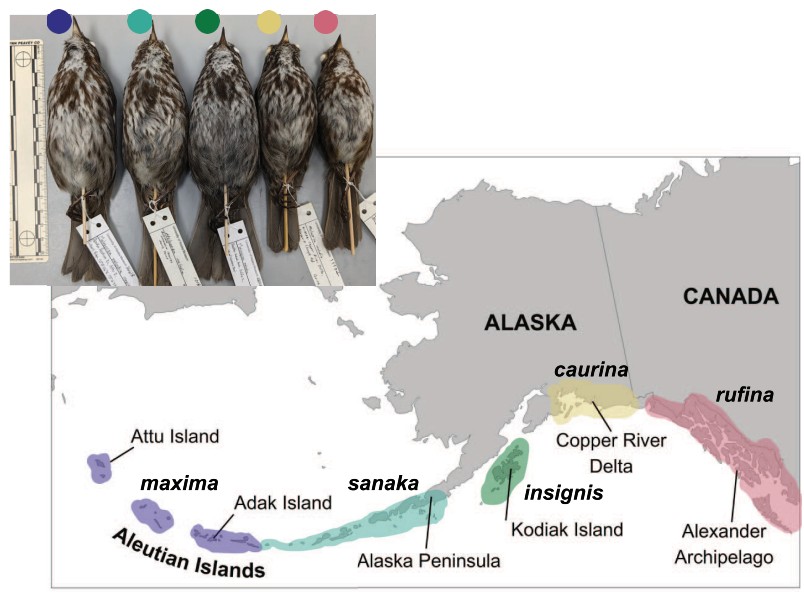

**Figure 1 Distribution of five subspecies of song sparrow (*Melospiza melodia*) in Alaska.** Insert at upper left is a photo of museum specimens of these subspecies. From west to east and left to right: *M. m. maxima* (dark blue), *M. m. sanaka* (teal), *M. m. insignis* (green), *M. m. caurina* (yellow), and *M. m. rufina* (pink). Photo credit: Caitlyn C. Oliver Brown.

marginal for continental subspecies, and are known to consume small marine invertebrates (*Gabrielson & Lincoln, 1959*; *Murie, 1959*; *Aldrich, 1984*). The subspecies that occur furthest east in Alaska (*caurina* and *rufina*) are seasonal migrants, residing largely on continental North America and migrate south, possibly to the Pacific Northwest, for the winter (*Aldrich, 1984*; *Patten & Pruett, 2009*). One subspecies (*insignis*) is primarily a year-round resident of the Kodiak Archipelago (*Patten & Pruett, 2009*). There are obvious phenotypic and natural history differences among these subspecies. The most notable phenotypic characteristics are differences in plumage coloration and body size: the most western subspecies in the Aleutian Islands are much larger than the eastern subspecies in southeast Alaska (Fig. 1; *Aldrich, 1984*; *Pruett & Winker, 2005b*; *Patten & Pruett, 2009*). These phenotypic differences suggest that these five subspecies have adapted to their local environments (*Kawecki & Ebert, 2004*; *Patten & Pruett, 2009*; *Winker, 2010*). We hypothesize that because *M. m. maxima* exhibits obvious phenotypic differences in plumage coloration, migration and dispersal, body size, and salt intake, candidate genes associated with these traits will be under selection.

Using museum specimens, we applied a combination of genomic and morphological analyses to study phenotypic differentiation among the subspecies of the Alaska song sparrow complex to examine the potential for adaptive evolution of high-latitude island populations. In so doing, we addressed the following questions: First, which morphological traits are significantly different among the five *M. melodia* subspecies in Alaska? Second, are candidate genes, associated with key phenotypic attributes, under positive selection in the most extreme variant, *M. m. maxima*? Third, what are the phylogenetic relationships among these subspecies?

## MATERIALS AND METHODS

### Phenotypic analyses

To examine phenotypic variation among the five *Melospiza melodia* subspecies, we obtained data from 227 vouchered specimens from the University of Alaska Museum (Table S1). We restricted our analyses to adult males with complete phenotypic and locality data because our sample size of adult females was not large enough to accurately quantify sexual dimorphism. Phenotypic data included: mass, wing chord length, tail length, tarsus length, bill length, bill height, bill width, and skull length (*Winker, 2000*). Measurements were taken to the nearest 0.1 millimeter (mm) or to 0.1 gram (g) for mass prior to specimen preparation (*Winker, 2000*). Data were reviewed and analyzed in R (v. 4.1.1; *R Core Team, 2021*) and RStudio (v. 2022.12.0.353; *Posit Team, 2025*) using the devtools v2.4.5 and tidyverse v.1.3.2 packages (*Wickham et al., 2022*, *2019*). We removed outliers that were three standard deviations away from the mean to account for human measurement error. In total, 28 outlier measurement values, out of 227 individuals, were removed. Phenotypic data were then visualized with boxplots and a principal components analysis (PCA) using the packages "ggplot2" (v.3.4.1; *Wickham et al., 2019*), "ggpubr" (v. 0.6.0; *Kassambara, 2023*), and "ggord" (v. 1.1.7; *Beck, 2022*). For the PCA, we first z-transformed the data and then used the R function "prcomp". When plotting the PCA, we used the first two principal components since they explained a majority of the variation. The 95% confidence intervals were calculated using the "ggord" package (*Beck, 2022*). ANOVA tests were used for each measurement and pairwise Tukey *post-hoc* tests were used to determine which subspecies pairs were different. R scripts and raw phenotypic data are available at https://github.com/coliverbrown/melospiza-melodia-phenotype. Portions of this text were previously published as part of a preprint (https://www.biorxiv.org/content/10.1101/2024.05.21.595201v1).

### Sampling and laboratory

For genomic analysis, we sampled high-quality vouchered muscle tissue samples from wild individuals archived at the University of Alaska Museum (Table S2). Our sample size included one individual from each of the five *M. melodia* subspecies that occur in Alaska. Two taxa were included as outgroups: the swamp sparrow (*M. georgiana*) and the Lincoln's sparrow (*M. lincolnii*). DNA extractions followed standard protocol for animal tissues using the Qiagen DNeasy Blood + Tissue Extraction Kit (Qiagen, Hilden, Germany).

Libraries were prepared using the iTru (Illumina dual-index) library protocols described in (*Glenn et al., 2019*). Briefly, we sheared the genomic DNA using a Bioruptor (Diagenode, Denville, NJ, USA) targeting a 500bp average fragment size. The sheared DNA was end-repaired, adenylated, and ligated to iTru adapters followed by limited-cycle polymerase chain reaction (PCR) of iTru primers to add indexes and complete the library molecules using Kapa Library Preparation Kit reagents (Kapa Biosystems, Roche, Basel, Switzerland). We sequenced the pooled libraries on an Illumina sequencer (Illumina, San Diego, CA, USA) to obtain paired-end (PE) 100 base reads. Raw sequence data have been deposited in NCBI Sequence Read Archive under BioProject PRJNA1114297.

## Bioinformatics

After sequencing, we constructed whole genome assemblies for each of the sequenced individuals. Our bioinformatics pipeline centered on the package PHYLUCE (*Faircloth, 2016*). Raw and untrimmed FASTQ data that contained low-quality bases were removed using Illumiprocessor (*Faircloth, 2013*), which incorporates Trimmomatic (*Bolger, Lohse & Usadel, 2014*). Raw paired end reads (read1, read2, and singleton files) were then mapped to a dark-eyed junco (*Junco hyemalis*; Passeriformes: Passerellidae) reference sequence (MLZ69236; *Friis et al., 2018*). Resulting sequences were indexed using BWA (*Li & Durbin, 2009*) and SAMtools (*Li et al., 2009*). Next, we used PICARD (*Broad Institute, 2019*) to clean the alignments, add read group header information, and remove PCR and sequencing duplicates. Single nucleotide polymorphisms (SNPs) were called for each individual against the reference, and Genome Analysis Toolkit (GATK, v.4.2.1; *McKenna et al., 2010*) was used to call and realign around indels, call and annotate SNPs, and filter SNPs around indels. We restricted the data to high-quality SNPs by adding a quality filter (Q30) before converting the resulting VCF file to a FASTA file using GATK.

## Selection analyses

To determine whether genes associated with phenotype were under positive selection, we identified candidate genes and tested for signs of selection in *M. m. maxima*. Candidate genes were chosen through a review of literature that identified genes that showed correlations with phenotypes of interest. We chose 37 candidate genes associated with body size (*Liu et al., 2015*), bill size and shape (*Lamichhaney et al., 2015*; *Chaves et al., 2016*; *Huang et al., 2022*), migration (*Vallone et al., 2007*; *Ruegg et al., 2014*; *Delmore et al., 2015*), dispersal (*San-Jose et al., 2023*), plumage coloration, including melanin and carotenoid pathway genes (*Walsh et al., 2012*; *San-Jose et al., 2017*), and salt tolerance (*Walsh et al., 2019*) (Table S3). We downloaded predicted coding DNA sequence data for the candidate from the annotated *Melospiza melodia melodia* genome accessioned on GenBank (*Rhie et al., 2021*; annotation name: GCF_035770615.1-RS_2024_02). We used these sequences (Table S3) to generate a local BLAST database and used BLASTn v2.12.0+ (*Altschul et al., 1990*; *Camacho et al., 2009*) to extract matching sequences from our *Melospiza melodia maxima* and *Melospiza georgiana* genome assemblies. Additionally, we extracted gene sequences from a *M. m. maxima* reference sequence (UAM31500; *Louha et al., 2020*). Using MAFFT v7 (*Katoh & Standley, 2013*), we manually generated multi-sequence alignments for each gene, removing indels to ensure that codon alignments were conserved.

We subsequently performed a McDonald-Kreitman test (*McDonald & Kreitman, 1991*) using a custom R script based on the algorithm described by (*Li & Fu, 2003*), which accounts for alternate mutation pathways for codons with more than one substitution difference (https://github.com/g33k5p34k/Melospiza_mktest). The McDonald-Kreitman test counts the number of synonymous ($D_s$) and non-synonymous ($D_n$) divergences between two species and synonymous ($P_s$) and non-synonymous ($P_n$) polymorphic sites within one of them (*McDonald & Kreitman, 1991*). In our case, we compared the polymorphic sites within *M. m. maxima* to divergences between *M. m.*

*maxima* and *M. georgiana*. These variables are used to calculate the neutrality index, $NI = (P_n/P_s)/(D_n/D_s)$. $NI > 1$ indicates an excess of amino acid polymorphisms, and $NI < 1$ is indicative of positive selection (*Rand & Kann, 1996*; *Stoletzki & Eyre-Walker, 2011*). Finally, a Fisher's exact test is calculated to determine whether the ratio of fixed substitutions between species and polymorphisms within species are significant.

### Phylogeny reconstruction

To reconstruct the phylogeny of the *M. melodia* complex and outgroups, we extracted ultraconserved elements (UCEs) from the whole-genome FASTA sequence files using PHYLUCE (*Faircloth, 2016*). We used UCEs because they are used to capture phylogenomic information from both shallow and deep time depths (*Faircloth et al., 2012*). We included the *Melospiza* individuals we sampled and the *J. hyemalis* reference sequence. Individual UCE loci were aligned using MUSCLE (*Edgar, 2004*) and were edge trimmed. We filtered this dataset to include loci that had all the *Melospiza* individuals and the *J. hyemalis* reference represented in a 95% matrix. The sequences were aligned by loci using MUSCLE and then concatenated into one dataset using PHYLUCE (*Edgar, 2004*; *Faircloth, 2016*). We obtained branch supports with the ultrafast bootstrap approximation (*Hoang et al., 2018*) implemented in IQ-TREE software (v.2.1.4; *Minh et al., 2020*). The phylogenetic tree was visualized and edited with FigTree (v.1.4.4; http://tree.bio.ed.ac.uk/software/Figtree/).

## RESULTS

### Phenotypic variation

Phenotypic variation among the subspecies was pronounced. Overall, subspecies that occur in the western range (*maxima, sanaka*, and *insignis*) were larger than eastern Alaska subspecies (*rufina* and *caurina*). *Melospiza melodia maxima* was approximately 1.6 times larger in body mass than *rufina* ($p < 0.0005$; Fig. 2; Table S4). For all other phenotypic measurements, differences were ~1.2 times larger (Fig. 2; Table S4). In all measurements, there was no significant difference between *rufina* and *caurina* (Fig. 2; Table S4). Additionally, *maxima* and *sanaka* showed significant differences only between bill height ($p = 0.007$) and bill width ($p = 0.006$) (Fig. 2; Table S4).

PCA analysis revealed that 55% of the variation among the five subspecies of *M. melodia* was explained in PC1 by five phenotypic traits: mass, wing chord, tail length, tarsus length, and skull length. Around 13% of the variation was explained in PC2 by the remaining three traits: bill length, bill width, and bill height (Fig. 3; Table S5). In the PCA graph, individuals tend to clump into two groups, mostly divided along PC1. The three western subspecies (*maxima, sanaka*, and *insignis*) group together, and the two remaining eastern subspecies (*caurina* and *rufina*) also group together (Fig. 3).

### Summary statistics of whole-genome sequencing

We obtained >400 million reads, ranging from 12,566,699 to 84,504,717 per individual (average = 60,119,612; Table S6). Coverage averaged 6.7×, ranging from 2.1× to 9.7× per individual (Table S6).

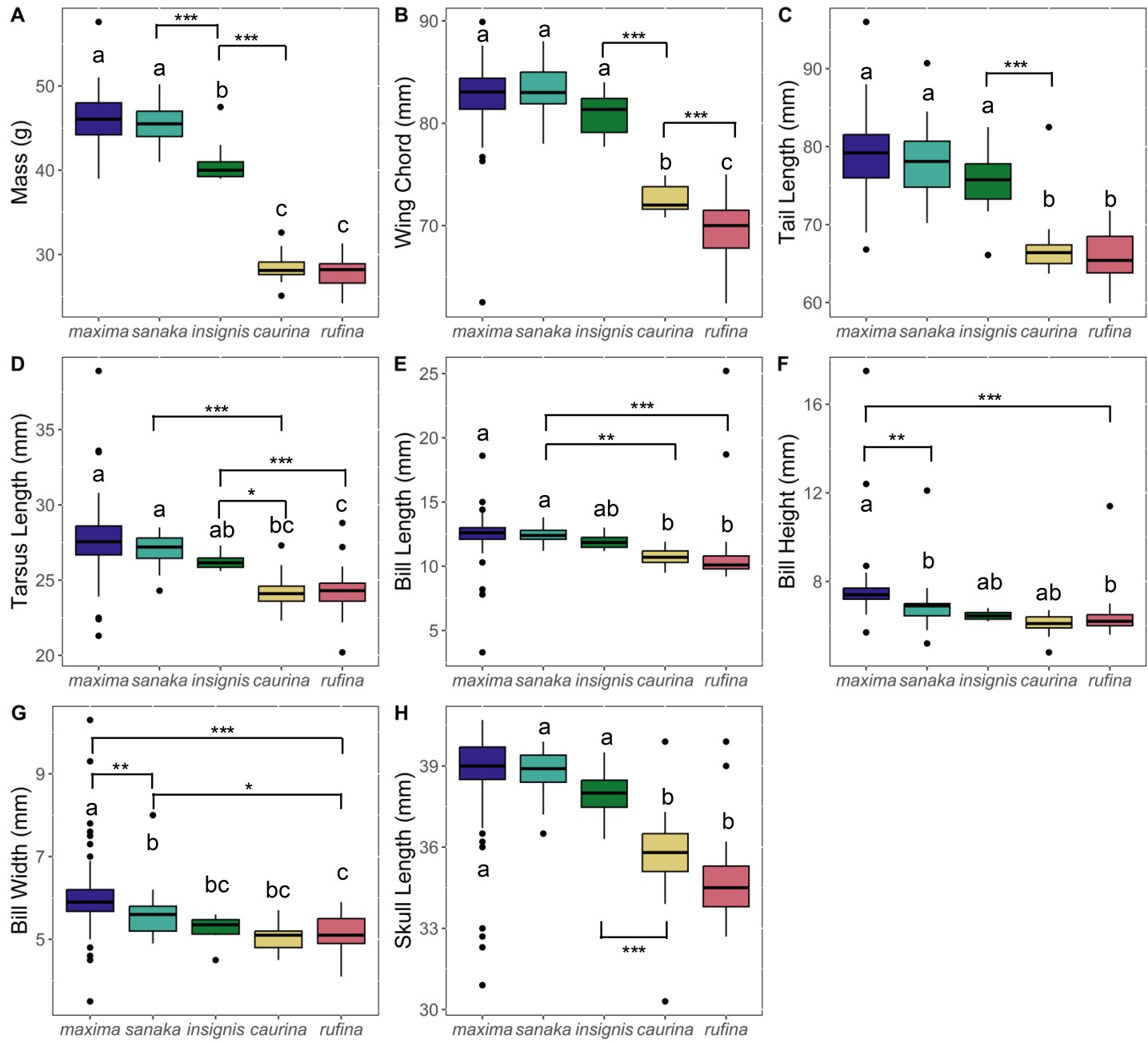

**Figure 2  Boxplots of phenotypic data for five *Melospiza melodia* subspecies:.** (A) Mass (g), (B) wing chord (mm), (C) tail length (mm), (D) tarsus length (mm), (E) bill length (mm), (F) bill height (mm), (G) bill width (mm), and (H) skull length (mm). Colors correspond to the colors in Fig. 1. Values with different letters are significantly different based on a Tukey's *post-hoc* test. ($p < 0.05$). Asterisks denote the level of significance in pairwise differences between subspecies for each trait: $^*p < 0.05$, $^{**}p < 0.01$, $^{***}p < 0.001$.           

## Selection analyses

Of the 37 candidate genes tested for signals of positive selection, two had significant differences between ratios of $P_n/P_s$ and $D_n/D_s$: *CD36* ($p = 0.026$) and *LAP3* ($p = 0.029$; Table 1). The neutrality index, which indicates direction and degree of deviation from

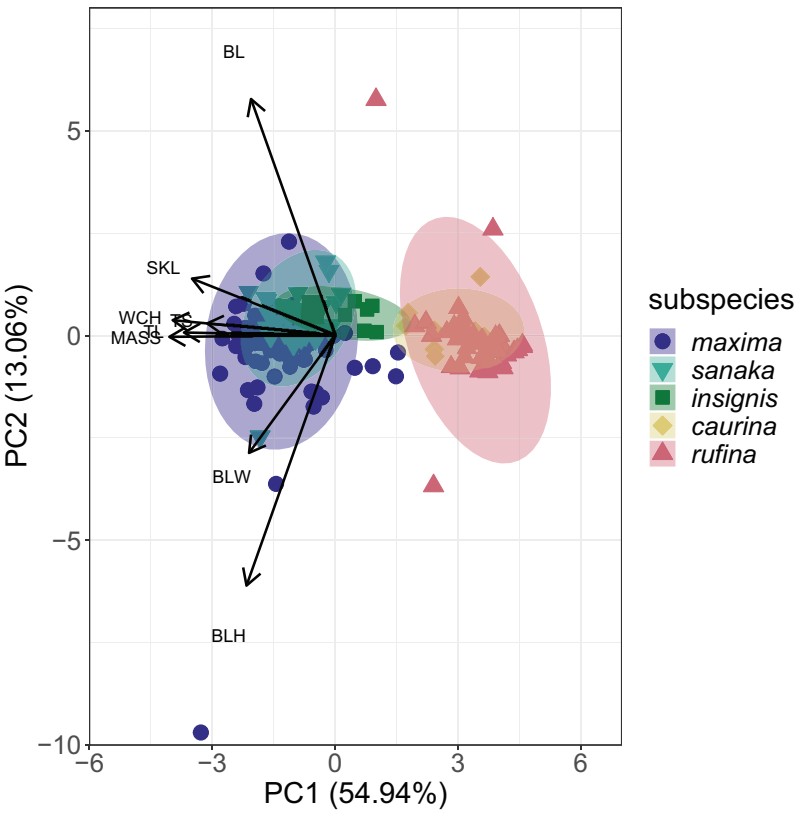

**Figure 3 PCA plot of phenotypic data for the five *Melospiza melodia* subspecies.** PC1 explains approximately 55% of the variation with five phenotypic traits: mass (MASS), wing chord (WCH), tail length (TL), tarsus length (TS), and skull length (SKL). PC2 explains 13% of the variation with the three remaining traits: bill length (BL), bill width (BLW), and bill height (BLH). Three subspecies (*maxima, sanaka*, and *insignis*) group together and are larger from the second group of two subspecies (*caurina* and *rufina*). Colors correspond to Figure 1.

neutrality, varied from 0 to 14.056, although 12 candidate genes had undefined neutrality indices, which occur when either $D_n$ or $P_s$ is zero, including *CD36* and *LAP3* (Table 1). Seven genes had a neutrality index < 1, which can be indicative of positive selection (*Rand & Kann, 1996*; *Stoletzki & Eyre-Walker, 2011*). Neutrality index values > 1 (seven genes) might indicate weak purifying selection, or the maintenance of slightly deleterious mutations in the population (*Rand & Kann, 1996*; *Stoletzki & Eyre-Walker, 2011*). Finally, for six genes, there were no differences between the coding sequences (Table 1, "NA").

## Phylogenetic reconstruction

After filtering UCE loci present in the *Melospiza* and *J. hyemalis* genomes to obtain a 95% matrix, the final alignment contained 4,793 loci. All nodes within the *M. melodia* clade received 100% support from the ultrafast bootstrap (Fig. 4). The phylogenetic tree shows that two of the western subspecies (*sanaka* and *insignis*) and the two eastern subspecies (*caurina* and *rufina*) form their own clades. However, *maxima* is a sister group to all other *M. melodia* subspecies (Fig. 4).

**Table 1 McDonald-Kreitman test results.**

| Phenotypic trait | Candidate gene | $D_s$ | $D_n$ | $P_s$ | $P_n$ | Neutrality index (NI) | Fisher's exact test $p$-value |
|---|---|---|---|---|---|---|---|
| Bill size | ALX1 | 4 | 0 | 1 | 0 | Undefined | 1 |
| | CCDC149 | 4.5 | 3.5 | 1 | 4 | 5.143 | 0.565 |
| | DLK1 | 3 | 1 | 9 | 2 | 0.667 | 1 |
| | HMGA2[†] | NA | NA | NA | NA | NA | NA |
| | LGI2[†] | NA | NA | NA | NA | NA | NA |
| Body size | AFAP1 | 5 | 2 | 5 | 3 | 1.5 | 1 |
| | **LAP3** | 8 | 0 | 4 | 5 | Undefined | **0.029** |
| | LCORL[†] | NA | NA | NA | NA | NA | NA |
| | QDPR | 2.333 | 5.667 | 2 | 8 | 1.647 | 1 |
| | SLIT2 | 10 | 1 | 12 | 2 | 1.667 | 1 |
| | WAPL | 7.667 | 0.33 | 18 | 11 | 14.056 | 0.076 |
| Dispersal | KCTD21[†] | NA | NA | NA | NA | NA | NA |
| | SLC2A1 | 4 | 0 | 10 | 0 | Undefined | 1 |
| | TGFB2 | 2 | 0 | 3 | 1 | Undefined | 1 |
| Migration | CLOCK | 2 | 3 | 3 | 1 | 0.222 | 0.524 |
| | CREB1[†] | NA | NA | NA | NA | NA | NA |
| | CRY1 | 4 | 0 | 3 | 0 | Undefined | 1 |
| | CRY2 | 1 | 0 | 3 | 1 | Undefined | 1 |
| | NPAS3 | 1 | 0 | 3 | 0 | Undefined | 1 |
| | PER2 | 15 | 4 | 9 | 0 | 0 | 0.273 |
| | PER3 | 7 | 2 | 15 | 4 | 0.933 | 1 |
| | YPEL1 | 1 | 0 | 1 | 0 | Undefined | 1 |
| Plumage color – carotenoid | APOD1 | 2 | 0 | 5 | 0 | Undefined | 1 |
| | BCO1 | 5 | 3 | 2 | 0 | 0 | 1 |
| | BCO2 | 2 | 3 | 14 | 10 | 0.476 | 0.632 |
| | **CD36** | 3 | 0 | 3.5 | 14.5 | Undefined | **0.026** |
| | HPS5 | 7 | 1 | 19.5 | 14.5 | 5.205 | 0.222 |
| | STARD3 | 4 | 1 | 4 | 0 | 0 | 1 |
| Plumage color – melanin | AGRP | 0 | 3 | 2 | 0 | 0 | 0.1 |
| | ASIP[†] | NA | NA | NA | NA | NA | NA |
| | MC1R | 3 | 1 | 2 | 2 | 3 | 1 |
| | MITF | 1 | 0 | 10 | 1 | Undefined | 1 |
| | PCSK2 | 2 | 0 | 3 | 0 | Undefined | 1 |
| | TYR | 3 | 2 | 6 | 3 | 0.750 | 1 |
| Salt tolerance | MMP17 | 1 | 1 | 9 | 2 | 0.222 | 0.423 |
| | MYOF | 13 | 2 | 12 | 4 | 2.167 | 0.654 |
| | WNK2 | 20.5 | 13.5 | 23 | 11 | 0.726 | 0.615 |

**Notes:**

The test calculates the total number of nonsilent ($P_n$) and silent ($P_s$) polymorphisms and nonsilent ($D_n$) and silent ($D_s$) divergences. The neutrality index (NI) is calculated as $(P_n/P_s)/(D_n/D_s)$. An undefined NI occurs when $D_n$ or $P_s$ is zero. Gene names and $p$-values in bold denote significant results (<0.05) from Fisher's exact test.

[†] No differences found between the three coding sequences tested.

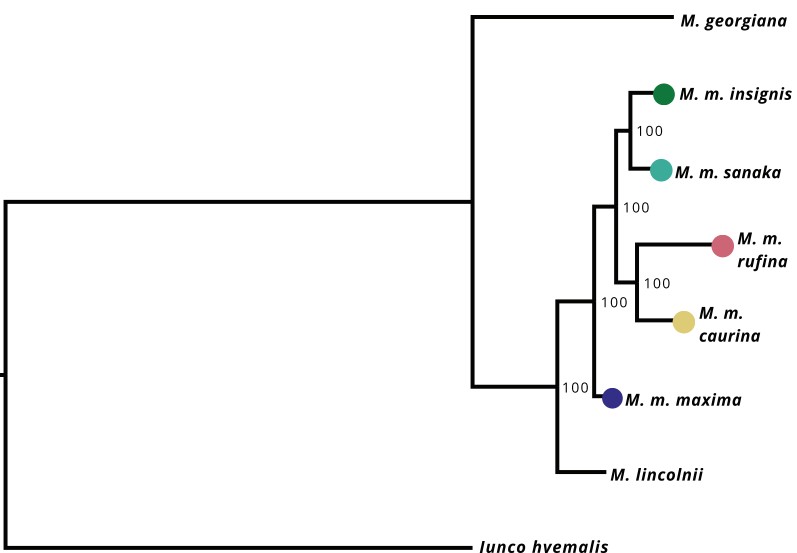

**Figure 4 Maximum-likelihood phylogeny based on 4,793 UCE loci.** Values on the nodes represent bootstrap values from 1,000 iterations. *M. m. maxima* is shown to be a sister group to the other *M. melodia* subspecies.

## DISCUSSION

Our results show that the westernmost subspecies of song sparrow (*M. m. maxima*) is approximately 1.6 times larger than eastern *rufina* (Fig. 2). The McDonald-Kreitman test showed that two candidate genes (*CD36* and *LAP3*) showed significantly different non-synonymous to synonymous substitution ratios between *M. m. maxima* and *M. georgiana*, indicating they are under selection, and the neutrality index suggested that seven more candidate genes were putatively under positive selection (Table 1). We reconstructed a novel phylogeny for the Alaska song sparrow complex, which places *M. m. maxima* outside the rest of the *M. melodia* clade (Fig. 4). This suggests that song sparrows colonized the Aleutian Islands early and underwent divergence before the eastern portion of the species' current Alaska range was recolonized.

### Phenotypic variation

The strong increase in body size among resident populations in the western Aleutians (Fig. 2) is concordant with previous research (*Aldrich, 1984*) and follows Bergmann's Rule, by which endothermic animals tend to be larger in cooler climates (*Bergmann, 1847*, *Meiri & Dayan, 2003*; *Carbeck et al., 2023*). When examining the PCA results (Fig. 3), the three western subspecies (*maxima, sanaka,* and *insignis*) grouped together and the remaining two eastern subspecies (*caurina* and *rufina*) grouped together. These groups are consistent with the different life-history strategies, with the larger, western subspecies being sedentary and the smaller, eastern subspecies being migratory. Sedentary birds tend to conform with Bergmann's rule more than migratory species because migratory species can seasonally escape more severe climatic conditions (*Meiri & Dayan, 2003*; but see *Ashton, 2002*). Likewise, island living could also have an impact on body size, because vertebrates

experience the "island rule", wherein large organisms on islands evolve toward smaller sizes and small organisms evolve toward larger sizes (*Lomolino, 2005*; *Losos & Ricklefs, 2009*). Many factors are hypothesized to contribute to this general trend, such as ecological release from predators and resource limitation (*Lomolino, 2005*). Other passerines that inhabit the Aleutian Islands, such as the gray-crowned rosy-finch (*Leucosticte tephrocotis*) and Pacific wren (*Troglodytes pacificus*), are also much larger in the western islands (*Murie, 1959*).

Of the western subspecies, *maxima* and *sanaka* were similar across all measurements except for bill height and bill width (Fig. 2; Table S4). *M. m. sanaka* had significantly narrower and shallower bills than *maxima*, which is consistent with previous research (*Patten & Pruett, 2009*; *Gibson & Withrow, 2015*). However, although slender bills are often associated with insectivory and stouter bills with granivory (*Aldrich, 1984*), little is known about the dietary differences of the two subspecies, so the ecological or evolutionary basis of this morphological difference remains unknown (*Aldrich, 1984*). Additionally, foraging behavior is not the sole cause of differences in bill size. Bird bills are thermoregulatory organs, facilitating heat transfer between the organism and the environment (*Tattersall, Arnaout & Symonds, 2017*; *Gamboa et al., 2024*). In the California Channel Islands, for instance, *Gamboa et al. (2024)* identified temperature as the main factor affecting differences in bill size in song sparrows between islands. Potentially, differences in temperature between the western Aleutian Islands and the Alaska Peninsula may cause the observed differences in bill size. Furthermore, local adaptation is not the sole cause of population differentiation, especially on islands. Founder effects have been shown to drive phenotypic and genetic differences between populations (*Kolbe et al., 2012*), and it is therefore possible that the differences we found between subspecies were neutral rather than adaptive, although we are unable to distinguish between the two in this study.

## McDonald-Kreitman tests identify potential genes under selection

Our McDonald-Kreitman tests identified two genes, *CD36* and *LAP3*, that showed significantly different non-synonymous to synonymous substitution ratios between *M. m. maxima* and *M. georgiana*, indicating that they are under selection (Table 1). This pattern was driven by the absence of fixed non-synonymous substitutions between species ($D_n = 0$), which suggests that these genes are under strong purifying selection ($NI$ = undefined). While none of the other candidate genes showed significant signatures of selection at a threshold of $\alpha = 0.05$, it is worth noting that our small sample size ($N = 2$ in-group and 1 out-group samples) likely led to severe underestimates of the number of within-species synonymous substitutions ($P_s$) relative to the other substitution types, thus reducing the power of the Fisher's Exact Test.

Despite our sample size limitations, it should nonetheless still be possible to identify putatively positively selected genes based on their NI values. Because the neutrality index can alternatively be expressed by the equation $NI = (P_n{}^*D_s)/(P_s{}^*D_n)$, our systematic underestimation of the number of $P_s$ sites should lead to higher than expected neutrality indices per gene, thereby underestimating the number of genes under positive selection.

Under our current sampling approach, loci with an *NI* < 1 can therefore be considered strong candidates for potential positive selection.

Because *M. m. maxima* exhibit obvious phenotypic differences from *M. georgiana*, we expected to observe directional selection in candidate genes associated with plumage coloration, migration and dispersal, body size, and salt intake. Although preliminary, our results generally align with these hypotheses, except for dispersal and body size-associated genes. For bill size, our analysis infers potential positive selection in *DLK1*, which has been shown to be associated with differences in bill morphology in Darwin's finches (*Chaves et al., 2016*) and Hawaiian honeycreepers (*Campana et al., 2020*). However, while *DLK1*-associated polymorphisms have generally been found in the transcription regulation region upstream of the gene in Darwin's finches, or at amino acids 100 and 244 in the Hawaiian honeycreepers, our results instead identify a novel substitution at amino acid 260, with a methionine residue in *M. georgiana* replaced by a threonine residue in *M. m. maxima*.

As for migration-associated genes, we found signals of positive selection in the *CLOCK* and *PER3* genes (Table 1), both of which are known to regulate avian circadian rhythms (*Kwak et al., 2024*). The *CLOCK* gene is particularly interesting, because other studies in birds have found that the number of glutamine residue repeats in the C-terminal domain appear to be linked to migration timing (*Bazzi et al., 2016*; *Saino et al., 2015*). However, we observed no glutamine repeat variation in our *M. m. maxima* or *M. georgiana* sequences and instead found two fixed amino acid substitutions at residues 6 and 221. As for the *PER3* gene, we found a single fixed substitution at amino acid residue 1,045, which occurs within the cryptochrome-binding region (*Kwak et al., 2024*). Our discovery of potential positive selection in these two migration-linked genes seems counterintuitive because *M. m. maxima* is sedentary whereas *M. georgiana* is migratory. However, it is possible that these polymorphisms could be linked to the suppression of migratory behavior in *maxima* individuals. Alternatively, these amino acid substitutions could be due to adaptations to the more variable photoperiods experienced by the *maxima* population at higher latitudes.

As for plumage coloration, we identified signals of potential positive selection in *BCO2*, which is associated with carotenoid coloration pathways (Table 1). *BCO2* has been a focus of carotenoid coloration studies in birds, most notably with regard to the increase carotenoid coloration in beaks, legs, and skin (*Walsh et al., 2012*; *Gazda et al., 2020*; *Enbody et al., 2021*). *Gazda et al. (2020)* found a single nonsynonymous mutation in *BCO2* exon 9, which disabled the enzyme's carotenoid cleaving ability, resulting in an increase in carotenoid accumulation in tissues. Additionally, in a study with Darwin's finches, *Enbody et al. (2021)* found a single nucleotide polymorphism associated with yellow *vs* pink beak coloration of nestlings. This polymorphism, found in *BCO2* exon 4, is synonymous but nonetheless alters the expression of *BCO2*, causing carotenoid deposition in bills (*Enbody et al., 2021*). However, neither of these mutations were observed in our data; instead, we found amino acid substitutions at residues 31, 87, and 311. While *M. m. maxima* does not possess obvious red coloration compared to other subspecies or *M. georgiana*, the number of fixed substitutions observed in our dataset nonetheless suggests that there may be differences in carotenoid deposition between the two taxa, which may explain the darker

phenotype of *maxima*. Another possibility is that changes in carotenoid deposition might be an adaptation by *maxima* to living in harsher environments. Studies indicate that increased carotenoid concentration stimulates the immune response of birds (*Pap et al., 2009*; *Toews, Hofmeister & Taylor, 2017*). In an experimental study with house sparrows, *Pap et al. (2009)* found that birds supplemented with carotenoids were able to fully compensate during parasitic infection. In humans, carotenoids also have antioxidant properties and prevent light damage to the retina (*von Lintig, 2010*; *Toews, Hofmeister & Taylor, 2017*). This combination of immune response and antioxidants might allow resident subspecies *M. m. maxima* to survive winters in the Aleutian Islands.

Additionally, one gene associated with the melanogenesis pathway, *TYR*, was identified to be under potential positive selection (Table 1). Positive selection in this gene is interesting because mutations at amino acid residues 246 and 371 have been found to cause melanin deficiencies in parrot plumage (*Ghosh Roy et al., 2025*). While our data do not show the same substitutions as *Ghosh Roy et al. (2025)*, one of our observed amino acid substitutions at position 369 is highly proximate to the H371Y substitution in the α8 helix of the psittaciform *TYR* protein, which lies close to the protein's active site. We also observed another amino acid substitution at residue 508, which lies in the transmembrane region of the protein. While we do not yet know how these mutations affect *TYR* enzymatic activity, we expect the positive selection of *TYR* to cause a higher melanin accumulation in *M. m. maxima* because of its overall darker coloration compared to *M. georgiana*. *M. melodia* generally follows Gloger's rule, or the tendency for birds in more humid environments to be darker in coloration (Patten & Pruett, 2010). More research into *TYR* and melanogenesis mechanisms is needed to determine why it is potentially under positive selection in *M. m. maxima*.

Finally, we found potential positive selection in two salt-tolerance candidate genes: *MMP17* and *WNK2* (Table 1). Neither of these genes have been studied extensively in birds. *MMP17* has been linked to drinking behavior and kidney function in mice (*Srichai et al., 2011*). In Nelson's sparrows, a region of elevated divergence in *MMP17* was noted between upland and salt marsh populations, suggesting potential adaptation to salt marsh habitats (*Walsh et al., 2019*). *WNK2* has been identified as an important component in osmoregulation, regulating cell volume in response to osmotic stress (*Kahle, Rinehart & Lifton, 2010*). Within birds, this region had an elevated divergence between upland and salt marsh savannah sparrows (*Walsh et al., 2019*). We assume that the inferred positive selection of both *MMP17* and *WNK2* are adaptations to the diet and life history of *M. m. maxima*. Individuals of this subspecies routinely forage along the marine beaches and rocky tidepools of the Aleutian Islands, consuming marine invertebrates (*Gabrielson & Lincoln, 1959*; *Murie, 1959*; *Aldrich, 1984*). Compared to *M. georgiana*, which consumes terrestrial invertebrates, we expect to see adaptations to a salty environment, similar to the phenomenon described by *Walsh et al. (2019)*.

Rather unexpectedly, we found no signal of positive selection in any of the seven candidate genes linked to differences in body size included in our study, despite *M. m. maxima* being on average 1.6 times larger than *M. georgiana*. This might be because other body size-linked genes could be responsible for mediating this phenotypic divergence.

Body size has been the focus of much research in the *Melospiza melodia* complex recently. *Carbeck et al. (2023)*, for instance, found nine genes associated with body size that showed divergence between *maxima* and *rufina* song sparrows and then validated these candidate gene findings by predicting the genotypes of five small song sparrow subspecies (*gouldii, heermanni, maxillaris, pusillula*, and *samuelis*) in California. Our study did not include any candidate genes found by *Carbeck et al. (2023)*, which should be further considered in future studies of song sparrows.

Additionally, we detected signals of strong negative or purifying selection ($NI$ = undefined) in twelve candidate genes, which was driven by the absence of fixed non-synonymous substitutions between species (Table 1). This presence of putative purifying selection in twelve candidate genes (*APOD1, AXL1, CD36, CRY1, CRY2, LAP3, MITF, NPAS3, PCSK2, SLC2A1, TGFB2,* and *YPEL1*) indicates these genes are conserved and potentially critical to the function of these organisms. It is also important to note that many genes are multi-functional and could contribute to functions outside of those we addressed here.

While these results generally align with our hypotheses, the McDonald-Kreitman (MK) test does have limitations. Modifications to the MK test exist that allow for selection to be detected in non-coding regions, but our application of the standard MK test means that we were unable to test for selection in regulatory regions surrounding protein-coding genes, which can be influenced by linked selection. Additionally, we were unable to account for the effect of slightly deleterious mutations, which can also make it difficult to identify adaptive evolution (*Andolfatto, 2008*). More importantly, small sample sizes and small effective population sizes could lead to elevated false positive rates (*Andolfatto, 2008*; *Subramanian, 2016*). However, although our analyses suffer from both issues (*Parsch, Zhang & Baines, 2009*), the fact that none of our candidate genes exhibited statistically significant levels of positive selection indicates that type I error was unlikely to have been a major issue. Limitations notwithstanding, our results provide a preliminary look into how *M. m. maxima* might have locally adapted to the Aleutian Islands. Future work should increase sample sizes to include more than two sequences tested, use coalescent simulations to determine significance levels (*Andolfatto, 2008*; *Subramanian, 2016*), and test more candidate genes (*e.g.*, *Carbeck et al., 2023*).

## Phylogenetic reconstruction

Our phylogenetic tree shows that the westernmost subspecies *maxima* is a sister taxon to all other Alaska *M. melodia* subspecies (Fig. 4). Further, the eastern Alaska subspecies *caurina* and *rufina* are grouped together and two of the western subspecies (*sanaka* and *insignis*) form another clade (Fig. 4). This suggests that *maxima* colonized the western Aleutian Islands first, underwent differentiation from other forms, and then at some later time, the eastern part of the Alaskan range was colonized by ancestors of the remaining subspecies. One possibility for this colonization history is that following an initial colonization of Alaska, the sedentary *maxima* survived in an Aleutian glacial refugium (*Pruett & Winker, 2005a*; *Winker et al., 2023*) while other Alaska populations were

extirpated, and then Alaska was re-colonized post-glacially. Seasonal migration likely aided this recolonization as glacial barriers retreated (*Pielou, 1991*), but some recolonizing lineages subsequently also became sedentary (*e.g.*, *sanaka* in the Aleutians). This hypothesis is inconsistent with the colonization history postulated by *Pruett & Winker (2005b)*, which suggested a probable postglacial stepwise process from east to west. Further sampling from across the species' range will be needed to confirm this hypothesis.

This is the first song sparrow phylogeny reconstructed from large sampling of the nuclear genome. Prior work using mitochondrial DNA (mtDNA) contradicts our results. *Zink & Dittmann (1993)* found no genetic structure in a mtDNA phylogeny across the species' range, and *Pruett & Winker (2010)* found no reciprocal monophyly or genetic structure within Alaska's subspecies. However, single-locus phylogenies are notoriously problematic because of issues like incomplete lineage sorting and gene flow (*e.g.*, *Avise & Wollenberg, 1997*; *Funk & Omland, 2003*; *Joseph, 2021*), and further work is needed to understand phylogenetic relationships among all the lineages in this species.

Interestingly, *maxima* and *sanaka* are similar in size, have similar life-history patterns, and occupy similar habitats. Yet, our phylogeny suggests *M. m. sanaka* is more closely related to all of the non-*maxima* subspecies. This phenotypic similarity could be due to convergent evolution in similar environments, or, more likely, due to adaptive alleles for Aleutian life introgressing through gene flow from *maxima* to *sanaka* (*Pruett & Winker, 2005b*; *Reding et al., 2008*; *Graham et al., 2021*).

It is possible that gene flow could be obscuring our phylogenetic results, however we do not expect this to be the case. Models show that moderate to high levels of gene flow can degrade phylogenetic signal and thus produce inaccurate phylogenetic trees (*Eckert & Carstens, 2008*; *Leaché et al., 2014*). Previous research by *Pruett & Winker (2005b)* found gene flow between *maxima* and *sanaka* to be at approximately 1.20–2.90 individuals per generation (Fig. S1). The exact amount of gene flow needed to obscure phylogeny is not known, but the current estimates for gene flow in the Aleutian Islands are not very high. Future research should determine whether phylogenetic relationships are being obscured by gene flow in this song sparrow complex.

## CONCLUSION

Among song sparrows in Alaska, phenotypic differences between subspecies suggest that selective pressures have acted heterogeneously on different populations. Our MK tests suggest that seven candidate genes associated with bill size, migration, plumage coloration, and salt tolerance might be under positive selection. This suggests that these traits may be under selective pressure in *maxima*, and lays the groundwork for future investigations with greater sampling depth and breadth. Additionally, we found signals of negative selection in twelve candidate genes, indicating these genes might be functionally critical and conserved rather than locally adaptive. The phylogenetic relationship that we recovered among Alaska song sparrow subspecies suggests a new hypothesis for the species' biogeographic history in this region. We suggest that ancestral populations of *M. m. maxima* colonized Alaska and the Aleutian Islands, and then persisted in a glacial refugium, likely while other

populations were extirpated. Then after a substantial period of time the rest of the state was recolonized by ancestors of the current subspecies east of *M. m. maxima*. This complex might be an ideal group for future studies of evolution of island populations and local adaptation.

## ACKNOWLEDGEMENTS

We thank Jacob Adams, Robin Andrews, Syndonia Bret-Harte, Stefanie Ickert-Bond, Rachel Richardson, Alex Sletten, Naoki Takebayashi, Laura Weingartner, Jeff Wells, and several anonymous reviewers for their invaluable input on earlier drafts. We also want to thank all the museum collectors who collected the specimens used in this project.

### Funding

This work was supported by the Kessel Fund, the UAMN Friends of Ornithology, and the Florida Institute of Technology. The funders had no role in study design, data collection and analysis, decision to publish, or preparation of the manuscript.

### Grant Disclosures

The following grant information was disclosed by the authors:
Kessel Fund, the UAMN Friends of Ornithology, and the Florida Institute of Technology.

### Competing Interests

K.A. Collier is employed by RENECO International Wildlife Consultants.

### Author Contributions

- Caitlyn C. Oliver Brown performed the experiments, analyzed the data, prepared figures and/or tables, authored or reviewed drafts of the article, and approved the final draft.
- Keiler A. Collier performed the experiments, analyzed the data, authored or reviewed drafts of the article, and approved the final draft.
- David J. X. Tan performed the experiments, analyzed the data, authored or reviewed drafts of the article, and approved the final draft.
- Kendall Mills performed the experiments, analyzed the data, authored or reviewed drafts of the article, and approved the final draft.
- Fern Spaulding performed the experiments, analyzed the data, authored or reviewed drafts of the article, and approved the final draft.
- Travis C. Glenn conceived and designed the experiments, authored or reviewed drafts of the article, generated sequence data, and approved the final draft.
- Christin Pruett conceived and designed the experiments, authored or reviewed drafts of the article, and approved the final draft.
- Kevin Winker conceived and designed the experiments, authored or reviewed drafts of the article, and approved the final draft.

## DNA Deposition

The following information was supplied regarding the deposition of DNA sequences:

The raw sequence data is available at NCBI Sequence Read Archive BioProject: PRJNA1114297.

## Data Availability

The raw data and code for phenotypic analyses are available at GitHub and Zenodo:

- https://github.com/coliverbrown/melospiza-melodia-phenotype.

- Caitlyn Oliver Brown. (2024). coliverbrown/melospiza-melodia-phenotype: *Melospiza melodia* morphology analysis (v1.0.0). Zenodo. https://doi.org/10.5281/zenodo.10994578.

## Supplemental Information

Supplemental information for this article can be found online at http://dx.doi.org/10.7717/peerj.19986#supplemental-information.

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
