# Peer review of "Evidence of positive selection and a novel phylogeny among five subspecies of song sparrow (Melospiza melodia) in Alaska"

_PeerJ, doi:10.7717/peerj.19986_

## Round 0.1 · original submission · Major Revisions

Dear Dr. Oliver Brown,

The reviewers of your paper agree that it is well-written. However, they all raised some points that you must please address.

all best,
Shaw Badenhorst

Reviewer 1 ·

Basic reporting

The manuscript is well-written and clear, with appropriate referencing and context. The figures and tables are appropriate and easy to understand; the code and data are accessible. The manuscript is self-contained with relevant results to the specific hypotheses being tested.

Experimental design

The research is original, the questions well defined, filling an important gap in our understanding of a particularly interesting group of subspecies. The work was performed to a high standard with sufficient detail to replicate.

Validity of the findings

Underlying code and data are provided. The approaches are sound and the conclusions are supported by the data.

Additional comments

The authors examine phenotypic and genetic differences among 5 subspecies of Aleutian Islands Song Sparrows to further our understanding of potential local adaptation in this group. Specifically, the authors document morphological traits that differ among the subspecies, test for evidence of selection on candidate genes associated with differences among subspecies (focused on the most divergent subspecies, maxima), and construct a phylogeny of the 5 subspecies using a subset of genomic data. The authors report interesting morphological divergence, evidence for selection on two candidate genes, and a well-resolved phylogeny that provides new insight into the likely colonization history of these islands.

Overall, the manuscript was clear and well-written with the conclusions supported by the data. The among-subspecies variation is amazing - I really appreciated the figures and tables. I had minor comments (below) requesting a little more information in a couple places.

Comments:

Generally, the work discounts the importance of non-adaptive evolution in the phenotypic divergence among subspecies, even though key evidence for fitness benefits of local vs nonlocal phenotypes is lacking (for good reason - it's hard to collect and beyond the scope of the study). That said, evidence that both adaptive and non-adaptive processes can underlie population differences (Kolbe, et al. 2012. Science 335, 1086-1089), particularly in islands, coupled with the lack of conclusive evidence for local adaptation (and for rejecting non-adaptive processes), make me feel that some sort of nod to non-adaptive processes (in introduction or discussion) would provide balance.

line 97: Related to the comment above, I would add 'potential for' between 'examine' and 'adaptive' to recognize that the phenotypic differences among populations could also have a non-adaptive explanation or contribution.

line 108: Why restrict to males? Are females similar? Is sexual dimorphism similar across populations? Please explain.

lines 248-250: "Birds and mammals experience the 'island rule', whereby large organisms on islands evolve toward smaller sizes and small organisms evolve toward larger sizes (Losos & Ricklefs, 2009)." Can you please add a sentence to explain why this might occur?

lines 262-274: How are carotenoids important in the divergent colouration of M. m. maxima? Are they depositing fewer carotenoids into their feathers/bare parts? Or could this reflect other trade-offs involving carotenoids (e.g., their use in immune function)? Overall, I wanted a stronger connection to your study subspecies here.

lines 283-284: "We suspect this gene remains under positive selection due to ancestral maxima individuals dispersing to the western Aleutian Islands." I don't really understand this explanation. I would expect that there would be positive selection for an allele that reduces dispersal/migration in M. m. maxima, consistent with the observed low gene flow among populations.

line 291: typo " Our study did not test for selection any candidate ..."

lines 296-306: Can you please add in a sentence to describe where the Aleutian subspecies fit into the broader Song Sparrow phylogeny (based of previous work)?

Reviewer 2 ·

Basic reporting

Brown et al. evaluate local adaptation to different environments among subspecies of Alaskan song sparrow. This group displays marked differentiation in a variety of phenotypic traits (notably body size) and presents a number of exciting opportunities to explore the role of local adaptation in shaping phenotypic and genotypic divergence. In general I thought the paper was well written, the figures are great and appropriate for the manuscript, and the analyses are well done. That said I do think there are a couple things the authors could do to better connect the described differences among groups with actual ecological differences. I think this will be essential if the authors want to make this a paper about local adaptation.

Experimental design

First, I would like to see some more discussion/justification about the candidate gene approach taken by the authors. Given that they invested in sequencing whole genomes for these species why did you choose to focus narrowly on 26 candidate loci? Why not explore a larger set of genes or take a more agnostic approach that evaluates signatures of positive selection across all annotated genes? I was also unclear why some candidate genes were selected. Specifically, why were carotenoid genes included, but not genes involved in the well-characterized melanogenesis pathway? Wouldn't color differences among song sparrow populations largely be due to melanin pigmentation and not carotenoids?

Second, I admire all the work that went into measuring specimens, but analyses primarily just describe differences among subspecies. While these analyses are totally appropriate, they do fall short of the local adaptation framework presented in the introduction. The authors could make a stronger case on the potential adaptive significance of these traits if they brought in some ecological data from WorldClim or other remote sensing databases. There is a lot of among individual variation even within subspecies in figure 3 and I would be curious if PC1 or PC2 correlate with certain metrics of temperature/precipitation, vegetation, or other ecological data?

Validity of the findings

All data provided.

Additional comments

Minor comments:
Line 86-88: can you add any available details on wintering range of these migratory subspecies?

Line 144-145: Why junco and not the available song sparrow genome (Louha et al. 2020)? Is it because Junco is chromosome level?

Line 170: Is 26 the number of the 34 candidates that passed the filters above?

Line 195-198: Did you look at how any of these traits may differ among populations after correcting for body size? For instance, migratory behavior could impact wing length and these differences may be more apparent if you account for body size differences among subspecies. This looks to be the case if you divide mean wing length values by body mass in table 1 (maxima: 82.7/47.2 = 1.7 vs. rufina: 69.7/27.9 = 2.5).

Line 234: delete "prior to"

Line 250: worth mentioning that other passerines also exhibit larger body sizes on the Aleutians (e.g. Savannah Sparrow, Gray-crowned Rosy-Finch).

Line 262: How many genes would you expect to exhibit a positive signature by chance? I.e. what is the probability of finding a false positive.

Line 303-306: This is a very interesting possibility, but I also think it should be noted that this study lacks sampling from outside Alaska that would be necessary to contextualize these results. For instance, is it possible that the Alaska subspecies are not monophyletic?

Figure 3 legend: length written twice for skull length.

Reviewer 3 ·

Basic reporting

N/A

Experimental design

This study examines the 5 subspecies of song sparrows along their Alaska range by targeting individual genes and testing for evidence of positive selection to test the hypothesis that island species would exhibit more local adaptation. The morphological variation that is observed within these 5 subspecies the study explains is probably caused by the geography of Alaska and the islands. The authors collected morphometric information from museum specimens already vouchered instead of taking measurements themselves. The study identifies two candidate genes that were significant and were related to dispersal and color. They also used tissue samples (n =1 per subspecies) to reconstruct a phylogeny. Based on their updated phylogeny, the study suggests that maxima are sister taxa with the other 4 subspecies, and they suggest a different colonization and dispersal pattern than previously described: the 4 subspecies having recolonized post-glacially instead of Pruett and Winker’s paper suggesting an east to west colonization. Overall, the paper is clear and concise, and the authors present clear objectives and have novel questions. Nevertheless, I do think some major improvements are needed primarily regarding key missing details within the methodology before publication. My recommendations are outlined below.

Major Comments:

86-87: how does migratory behavior tie into the conclusion of the new phylogeny?

100: I think the candidate genes need to be introduced before stating the hypotheses/questions. How does this help accomplish the bigger picture? How do other studies use candidate genes to understand local adaptation within subspecies?

108-109: I understand that the sampling of this study was dependent on the individuals available as vouchers specimens. However, I think additional justification needs to be added as to why only males were included. Were there not enough females in the collection to have a 50:50 sampling dataset? If so, this should be added as the justification. Are there known size differences between the two sexes and if so, how would that impact the study?

114-115: In Table S1, the name of the preparator is given in the data so I am wondering
what is the justification for using the three standard deviations from the mean method to account for human error rather than using the preparator as a random variable in the stats since a person’s measuring error, in this case, will be assumed to be consistent by individual. As it is now, the stats only accounts for overall outliers in the measurements and does not account for the different individuals that conducted the measurements at the time the specimens were prepared.

114: Continued: The number of outliers removed should be listed – maybe this could be added to the supplement.

116: Many details regarding how the PCA analysis was conducted appears to be missing. How were the number of clusters chosen following what methods? How were the number of principal components selected and was this based on how much they explained the variation of the data?

123: What kind of tissue samples were used and was this consistent across individuals?

156-159. What led the authors to select the candidate genes that they selected? What were the “phenotypes” of interest. More details are needed here as to why they were selected. While mentioned a few times, I never noticed anything about the salt tolerance and the purpose for testing this. How does life history differences play a role with the candidate gene questions from what is known? Brief introduction of this could be added after lines 93-94.

192-196: The adjusted p-values after the post-hoc test should be added in the table or in the text somewhere for reference.

201: The percent variance of each variable included in the PCA should be added as well as the eigenvalues and individual variable weights. This could be included as a separate table.

Discussion:

286-287: why did the authors expect to see more candidate genes under selection including the salt tolerance one? Again, I think more details are needed in the introduction of the paper to set this up earlier.

Validity of the findings

N/A

Additional comments

Minor Comments:

94: Are there any other additional studies that could also be cited other than Winker that supports that these phenotypic differences suggest local adaptation to the environment is occurring?

164: Was this supposed to be “M. georgiana” and Lincoln’s sparrow (not maxima?) since that is the second outgroup species listed earlier?

270-274: I think the ending to this paragraph can be cleaned up a bit. It is not clear how the color gene plays a role within the conclusions of this study. How might carotenoids be important and a driver of selection in this case? What is known about carotenoids in song sparrows or closely related sparrows? Carotenoids can be important antioxidant sources so is it possible that individuals that migrate versus the sedentary ones have more antioxidants due to migratory status? Or cold acclimation for the ones that experience lower temperatures or temperature fluctuations? While it might be beyond the scope of this study, it would be interesting to quantify carotenoids differences in the plumages of these birds although with their brown and white plumages, carotenoids might not play an important role for plumage, especially since this species does not exhibit clear sexual dichromatism pattern for mate selection.

290: Of these 5 small song sparrows in CA that Carbeck et al. 2023 tested, what subspecies did they belong to? It would be good to list that here.

310: Does sanaka and maxima experience similar environments relative to the other subspecies?

312. after “through” insert “to sanaka” or reword so it is not a run-on sentence.

327: Add the word “patterns” or something similar after the statement “carotenoid breakdown and dispersal” since without it, it sounds like it is discussing the dispersal of carotenoids in tissues rather than juvenile dispersal.

---

## Round 0.2 · Minor Revisions

The reviewers are generally satisfied with the changes made, but requested some final, minor changes to improve the paper.

Reviewer 1 ·

Basic reporting

The ms meets all of the relevant criteria for publication.

Experimental design

The ms meets all of the relevant criteria for publication.

Validity of the findings

The ms meets all of the relevant criteria for publication.

Additional comments

line 478: " ... indicating these genes might be functionally critical and conserved rather than adaptive."
If the genes are functionally critical and conserved, then they are adaptive. Either delete 'rather than adaptive' or replace with 'rather than locally adaptive."

Overall, I really like the changes made to the ms. The ms is a valuable and well-written contribution to the literature.

Reviewer 2 ·

Basic reporting

I previously reviewed this article (reviewer 2). The revised manuscript addresses many of the concerns raised by myself and other reviewers. In particular, I appreciate the inclusion of candidate genes from the melanogenesis pathway.

The authors did push back against one of my comments and seemed to mis-understand what I was requesting. My suggestion to include WorldClim data was intended to understand what environmental variables influence morphological variation not genomic variation. I think this would still be a good idea as it may help explain bill size differences among populations.

Some minor comments that would help improve the final manuscript.
Lines 284-287: bird bills have many more functions than just diet that could also be shaping differences between these subspecies. See Tattersall et al. 2017 for a good reference. Again this is an example of where connecting environmental variation to morphological variation would help explain some of the patterns reported in this study.
Tattersall, G.J., Arnaout, B. and Symonds, M.R.E. (2017), The evolution of the avian bill as a thermoregulatory organ. Biol Rev, 92: 1630-1656. https://doi.org/10.1111/brv.12299

Lines 354-356: I am very skeptical of this explanation. However, I do like the subsequent discussion of alternative functions of carotenoids beyond color. I would also note that many genes are multi-functional and may contribute to functions outside the candidate pathway focused on in this study. For instance, CD36 has many functions beyond carotenoid processing: https://www.genecards.org/cgi-bin/carddisp.pl?gene=CD36
It would be valuable to add the additional caveat that these genes under selection could be doing things outside their hypothesized function.

Experimental design

all comments made under basic reporting

Validity of the findings

all comments made under basic reporting

---

## Round 0.3 · accepted · Accept

The authors adequately addressed the last set of minor comments from the reviewers.